# Design of miniprotein inhibitors targeting complement C9 to block membrane attack complex assembly

Min Li[1,2,8], Ningning Wang [3,8], Xiaoyan Fu[1,8], Gege Wei[1,8], Ze Zhang [4,8], Yanghan Yu[1], Tianshui Xue[5], Yifei Zhao[6], Jinheng Pan[6], Dongfeng Wang[1], Meifang Liu[1], Yong Li[7], Jinbao Tang[5], Longxing Cao [4], Zhaocheng Jian[2], Shujuan Liang [1] ✉ & Bowen Yu [1,2,4] ✉

The abnormal formation of the membrane attack complex (MAC) is intrinsically linked to a range of acute and chronic immune diseases. The insertion of complement C9 into the membrane is the final step and kinetic bottleneck of MAC formation. However, research on blocking the MAC formation of C9 is currently limited. Given its broad, flat, and polar functional interface, complement C9 is a challenging target for rational design. Here, we utilize deep learning-based methods for protein scaffold generation, sequence design, and complex structure prediction to de novo design mini-protein inhibitors that specifically block the membrane insertion of soluble complement C9. The binding affinity of the mini-protein inhibitor is further optimized to 700 pM via partial diffusion. Design accuracy and binding specificity are verified through X-ray crystallography and biochemical studies. An in vivo acute hemolysis inhibition assay reveals that the C9 mini-protein inhibitors remain effective against hemolysis even 8 minutes after complement activation, outperforming the complement C5 inhibitor eculizumab. The de novo designed C9 mini-protein inhibitors can offer an optional therapeutic approach for the prevention and treatment of acute or chronic immune diseases associated with abnormal complement activation.

The complement system, ancient and evolutionarily conserved, is one of the most important components of the immune system. It defends against invading pathogens while preserving host homeostasis. Regardless of whether the initiating trigger is the classical, lectin, or alternative pathway, all three activation pathways converge on the formation of the membrane-attack complex (MAC). Aberrant MAC assembly on host cells underlies a spectrum of autoimmune disorders, including paroxysmal nocturnal hemoglobinuria (PNH)[1,2], atypical hemolytic uremic syndrome[3], myasthenia gravis[4,5], and neuromyelitis optica spectrum disorders[6,7].

Complement C5 cleavage into C5a and C5b initiates MAC formation. C5b rapidly binds C6 to form the C5b6 complex, which

[1]Key Laboratory of Immune Microenvironment and Inflammatory Disease Research in Universities of Shandong Province, School of Basic Medical Sciences, Shandong Second Medical University, Weifang, China. [2]Interventional Vascular Surgery Center, Affiliated Hospital of Shandong Second Medical University, Weifang, China. [3]First Affiliated Hospital, Weifang People's Hospital, Shandong Second Medical University, Weifang, China. [4]School of Life Sciences, Westlake University, Hangzhou, China. [5]Department of Biochemical Drugs, School of Pharmacy, Shandong Second Medical University, Weifang, China. [6]Biomedical Research Core Facilities, Westlake University, Hangzhou, China. [7]Department of Oncology, Guizhou Provincial People's Hospital, Guiyang, Guizhou, China. [8]These authors contributed equally: Min Li, Ningning Wang, Xiaoyan Fu, Gege Wei, Ze Zhang. ✉e-mail: liangshj@sdsmu.edu.cn; yubowen@sdsmu.edu.cn

sequentially recruits C7, C8, and multiple C9 molecules. C6, C7, C8α, C8β, and C9 undergo a conserved conformational switch that exposes a shared pore-forming domain, ultimately assembling into the membrane-attack complex (MAC) on the target membrane[8–11]. Eculizumab, a monoclonal antibody that prevents cleavage of C5 into C5a and C5b, represents a landmark in complement therapeutics[12–20]. However, blockade of C5 also inhibits the release of C5a, which may increase infection risk, and some patients are insensitive to treatment with eculizumab due to the residual C5 activity[21] and genetic polymorphisms of C5[22]. Recent studies have therefore evaluated antibodies targeting C6[23] or C7[24], demonstrating that downstream inhibition can achieve equivalent or superior hemoprotection without compromising C5a-mediated immunity.

The activation of the complement system involves a series of enzymatic cascade reactions. Once initiated, the enzymatic cascade assembles the MAC in a relatively short period of time to attack the target and the inhibitors such as eculizumab are kinetically disadvantaged. In vitro, target cell lysis can be completed within 5 min[23]. Indeed, incorporation of the first C9 is the kinetic bottleneck of MAC formation, thereafter 18 soluble C9 molecules rapidly oligomerize to complete the pore[25]. In fact, the widely expressed GPI-anchored glycoprotein CD59 could sterically block the insertion of the first C9 β-barrel hairpin into the membrane, preventing C9 polymerization and membrane-attack complex (MAC) formation[11]. Loss of the GPI anchor leaves CD59 unable to reach the membrane, depriving red cells of their final line of defense against complement and representing a central mechanism underlying the chronic hemolysis of PNH[26]. Besides, the constitutive expression of CD59 is insufficient to prevent massive MAC formation in hyperacute settings such as acute hemolytic transfusion reaction (AHTR)[27,28], paroxysmal cold hemoglobinuria (PCH)[29], or cold agglutinin disease (CAD)[30–32]. Consequently, designing protein to direct blockade of C9 insertion may act as a "last-minute brake" even after cascade initiation, providing life-saving potential for both chronic and acute hemolysis. Despite this therapeutic rationale, C9-directed inhibitors are rarely reported. An early report showed that a commercially available anti-C9 monoclonal antibody X197 conferred inhibitory activity in a non-standard hemolysis-inhibition assay[33]. Separately, the O1-antigen of *Klebsiella* LPS was found to block C9 polymerization[34]. However, neither study evaluated the therapeutic potential of these inhibitors for hemolytic disorders. The key polymerization interface of C9 is highly conserved across mammals, which increases the difficulty in immunological screening of functional monoclonal antibodies. Moreover, the relatively high serum concentration of C9 requires C9 inhibitors have high doses and exquisite functional inhibition specificity.

With the rapid development of AI-based diffusion denoising technology for generating target binders, we can more accurately generate binding proteins for a broader range of target types[35,36]. This technological advancement has to some extent compensated for the inherent shortcomings of antibody based screens that targets conserved function-critical sites. Additionally, it allows the production of high-affinity mini-proteins through low-throughput experimental workflows. Moreover, mini-protein binders can be produced in *E. coli* under simple culture conditions, with high expression levels and thermal stability[37]. Compared with the higher experimental barriers and greater production costs associated with antibody production, mini-protein binders hold vast potential for both research and pharmaceutical applications. Owing to strict structure-guided design, high affinity mini-protein binders have been reported for numerous ligands such as TNFR[36], EGFR[37], IL-1β[38], and snake venom toxins[39]; nevertheless, the design of high-affinity and functional inhibitors against complement components, especially the structurally unique C9 protein remains challenging.

In this work, we utilize deep learning-based methods to de novo design mini-protein inhibitors that specifically block the membrane insertion of soluble complement C9. Screening of 103 candidates through an in vitro hemolysis inhibition assay identified the best-performing scaffold. The binding affinity of the mini-protein inhibitor is further optimized to 700 pM via partial diffusion. Design accuracy and binding specificity are verified through X-ray crystallography and biochemical studies. An in vivo acute hemolysis inhibition assay reveals that the C9 mini-protein inhibitors outperform the complement C5 inhibitor eculizumab.

## Results

### Design of mini-protein binder target C9 with RFdiffusion

As the monomeric structure of human complement C9 has not yet been experimentally determined, we employed the crystal structure of monomeric mouse C9 as our initial design template. C9 adopts a disc-shaped structure (Fig. 1a, b), and its transition from soluble monomer to membrane-inserted pore relies on an extensive 77 × 50 Å interface (Fig. 1b). This large interaction region is flat and contains few separated surface hydrophobic residues (Fig. 1a, b). Theoretically, binding to this region of soluble C9 can effectively inhibit its polymerization (Fig. 1a). Since binder design favors surface-exposed hydrophobic residues as hotspots[35,36], we carefully checked the C9 interaction region for spatially clustered, interface-critical hydrophobic residues and grouped them into several sets. Two independent hydrophobic hotspots were initially chosen for RFdiffusion scaffolding: Site-1 (V196, I199, V305) and Site-2 (L217, A233, F235) (Fig. 1b). For each site, 1500 backbones (50-80 residues) were generated; three sequences per backbone were designed using ProteinMPNN and scored initially with AlphaFold2 (Supplementary Fig. 1 & 2a). In the scoring scheme, pae_interaction represents the predicted aligned error that AlphaFold2 assigns to the binder–target interface. It is the key metric for assessing how confident the model is in the interface prediction: the lower the value, the more reliable the prediction and the better the spatial fit between binder and target. Designs with pae_interaction <10 are generally considered acceptable. Complementarily, plddt_binder measures the per-residue confidence of the binder alone. A high value indicates that the sequence is likely to fold into a stable, well-defined monomeric structure; a threshold of > 90 is commonly required[35,36]. After applying relaxed filters (pae_interaction <15, plddt_binder >88), Site-1 achieved a 1.20 % success rate, while Site-2 approached 0 % (Supplementary Fig. 2a). In the second round of calculation, we introduced a new site, Site-3 (F231, M476, P478, Y480, I505) for calculation (Fig. 1b), and generated 4,300 scaffolds. The initial success rate for Site-3 was 0.16% (Supplementary Fig. 2b). We further combined Site-1 and Site-3 to create a new hotspot combination, Site-1 + 3 (V196, I199, V305, P478, Y480, I505) (Fig. 1b), and generated 8000 scaffolds with lengths ranging from 80 to 130 residues to generate larger binders, achieving an initial success rate of 1.18% (Supplementary Fig. 2b). In a final round, the diffuser.T was set to 120, Site-3 residues were refined to P478, Y480, L485, I505, and a total of 14,000 backbones were sampled across Site-1, Site-3 and Site-1 + 3, yielding success rates of 1.57%, 0.25% and 2.72%, respectively (Supplementary Fig. 2c). After filtering the combined ensemble (pae_interaction <8.5, plddt_binder >90), 103 top-ranking mini-protein binders that target three distinct binding sites were selected for experimental validation (Fig. 1c).

### Initial screening of mini-protein inhibitors for complement C9

The predicted monomeric human C9 structure aligns closely with the mouse C9 crystal structure, suggesting the feasibility of screening for a mini-protein inhibitor that can inhibit C9 polymerization across mouse and human (Fig. 2a). 103 mini-protein binders were expressed in *E. coli* and purified by affinity chromatography. Unlike traditional in vitro protein–protein interaction screens, we directly evaluated C9 mini-protein inhibitors using a functional hemolysis inhibition assay with mouse serum and sheep red blood cells. The mini-protein inhibitors were tested at a fixed concentration of 40 μM for initial screening. The

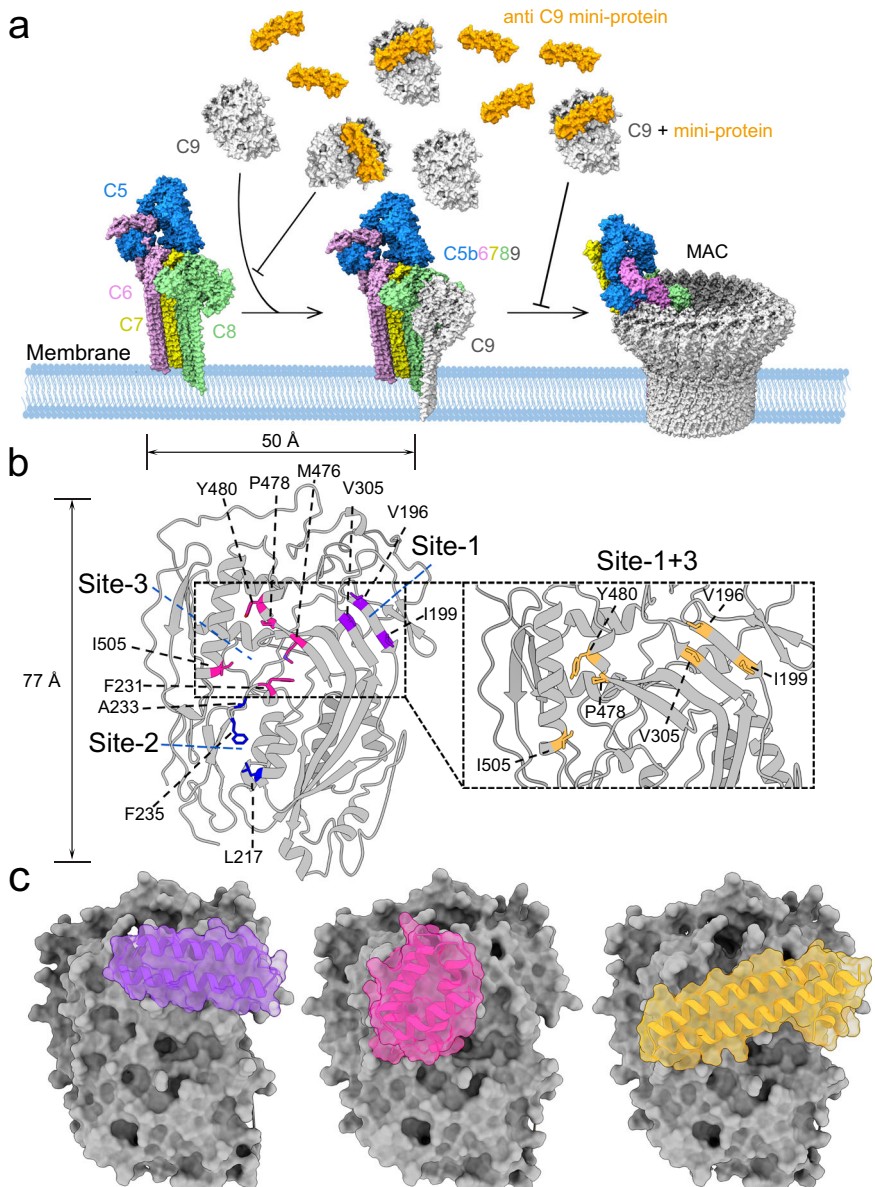

**Fig. 1 | Computational design of complement C9 mini-protein inhibitors.**
**a** Schematic overview of the MAC formation and the designed block mechanism by mini-protein inhibitors targeting complement C9. Color scheme: complement C5, blue; complement C6, pink; complement C7, yellow; complement C8, green; complement C9, gray; and complement C9 mini-protein inhibitor, orange. All MAC assemblies: PDB ID: 6H03; Soluble forms of complement proteins are derived from C9: PDB ID: 6CXO. **b** Complement C9 depends on a huge and flat interface to perform membrane insertion function. Site-1 is shown in purple, Site-2 in blue, and Site-3 in pink. Zoom-in view shows Site-1 + 3 (orange). **c** Structural snapshots of the mini-protein inhibitors selected for experimental screening that target three distinct hotspots on C9, Site-1 (Binder-87, purple); Site-3 (Binder-21, pink); and Site-1 + 3 (binder-47, orange). Design models for **c** are provided in zenodo.org with accession code: 18530064.

results showed that at a low serum concentration of 10%, five binders (18, 47, 50, 69, and 87) exhibited relatively good inhibitory effects (Fig. 2b). When the serum concentration was raised to 15 %, only Binder-47 and 87 retained potent inhibitory activity, with Binder-47 being more effective (Fig. 2c). Binder-47 revealed obvious dose-dependent hemolysis inhibition from 1.85% to 96.31% in a concentration range of 10 nM to 10 μM (Fig. 2d, e), suggesting a potential specific inhibition targeting complement C9 membrane insertion. Binder-47 was designed from the hotspot Site-1 + 3 scaffolding, which contain a relatively larger interface. This interface was highly conserved across mammals (Fig. 2a & Supplementary Fig. 3), only three contact residues were different between human and mouse C9 (Fig. 2a & Supplementary Fig. 3). Considering the potentially higher accuracy of AlphaFold3 in modeling protein–protein complexes, we used AlphaFold3 to

predict the complex structure of Binder-47 with human C9. AlphaFold3 modeling predicted that Binder-47 also well interacts with human C9 (Fig. 2a), indicating that it may serve as a cross-species inhibitor. Consistently, Binder-47 also inhibited hemolysis mediated by human serum (Fig. 2d, e). Biolayer interferometry (BLI) analysis using recombinant Fc-tagged complement C9 proteins revealed that Binder-47 interacts more strongly with mouse C9, with $K_d$ values of 3 and 22 nM (Fig. 2f), respectively. These results are consistent with the hemolysis inhibition assay (Fig. 2e).

**Affinity maturation using partial diffusion**
Hemolysis inhibition assays indicated that the scaffold of Binder-47 had strong optimization potential for complement C9 mediated MAC formation inhibition. We focused on optimizing the inhibitory effect of

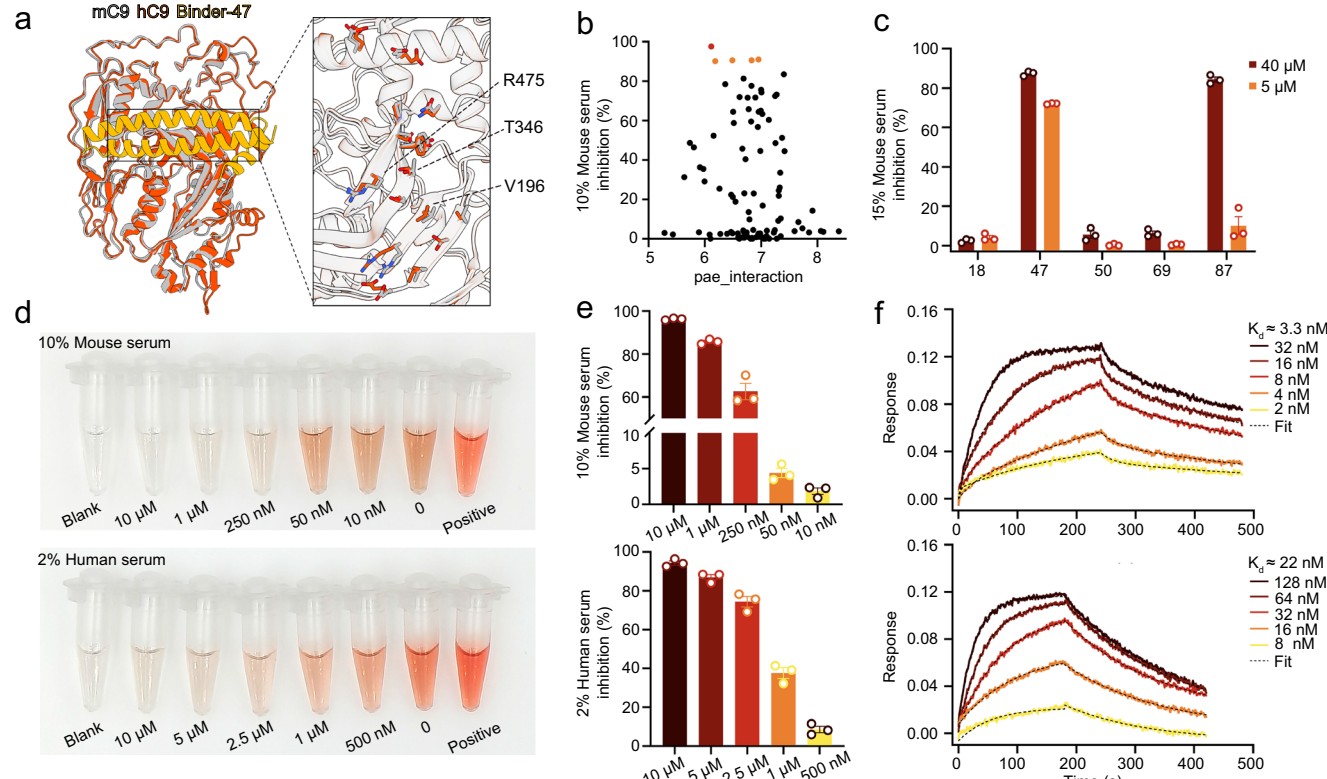

**Fig. 2 | Initial screening of mini-protein inhibitors for complement C9.**
**a** Complex structure model of the human complement C9 and Binder-47 predicted by AlphaFold3 (pTM = 0.85, ipTM = 0.93, indicating a highly reliable binding interface). The mouse complement C9 crystal structure was aligned onto the human complement C9. Zoom-in view of three different interface residues (V196, T346, and R475 represent residues in mouse C9). Color scheme: mouse complement C9 (mC9), gray; human complement C9 (hC9), red; Binder-47, orange. PDB ID: mouse complement C9: 6CXO. **b** In vitro hemolysis inhibition rate plotted against pae_interaction scores. Orange dots denote binders with inhibition rates above 90%; the red symbol marks the top-performing binder (Binder-47). **c** The inhibition effect of five identified binders against 15% mouse serum at 5 or 40 μM. **d** Photographs of Binder-47 hemolysis inhibition effect to mouse serum (top) and NHS (bottom). **(e)** The corresponding hemolysis inhibition rate in (**d**). **f** BLI characterization of the interaction of mini-protein Binder-47 with both mouse C9 (top) and human complement C9 (bottom). Graphs in (**c**, **e**) show mean ± SEM of n = 3 biologically independent samples. Source data for are provided as a Source Data file (Predicted model for (**a**) is provided in zenodo.org with accession code: 18530064).

Binder-47 on human complement C9 function, which has greater pharmaceutical value compared with mouse derived C9. The complex structure of Binder-47 and human C9 was predicted by AlphaFold3 and further used as input for partial diffusion (Fig. 2a & Supplementary Fig. 1). In contrast to previous reports that fixed diffuser.partial_T at 15 or 25[36], we sampled a broader range (1-25, step = 1), generating 25,000 backbones. Generated backbones were applied to ProteinMPNN sequence design and ranked by AlphaFold2 prediction scores. Applying stricter cut-offs (pae_interaction <6.2, plddt_binder > 90) yielded 60 top-ranking sequences for gene synthesis. We first fixed the mini-protein inhibitors concentration at 500 nM using human serum in hemolysis inhibition assay and identified 30 candidates that outperformed the parental Binder-47 (Fig. 3a & Supplementary Data 1). Dose responsive assay narrowed the list to six binders (P9, P14, P19, P39, P45 and P57), with P9 and P57 performing best. At the concentration of 12.5 nM, P9 and P57 achieved hemolysis inhibition rates of 98.21% and 55.92%, respectively (Fig. 3b), after setting a sufficiently low pae_interaction score for screening, the hemolysis inhibition rates obtained from experimental validation showed no strong correlation with the pae_interaction similar to initial screening, indicating the importance of experimental validation (Figs. 2b and 3a). Structural comparison revealed the Cα r.m.s.d values of P9 and P57 relative to Binder-47 were 1.01 and 0.875 Å, suggesting a slight adjustment of scaffold during partial diffusion (Fig. 3c). The residues involved in interface interactions also exhibited diversity in P9 and P57 (Fig. 3c). Parallel screening using P1-P60 against mouse serum (Supplementary

Fig. 4a & Supplementary Data 1) confirmed P57 as the most potent cross-reactive binder, reaching an 80% inhibition rate at a final concentration of 100 nM, exceeding the parental Binder-47 (Fig. 3d). BLI results indicated an obvious enhancement of affinity with a $K_d$ of 0.7 nM between P9 and human complement C9 (Figs. 3e) and 1.3 nM for P57 and mouse complement C9 (Fig. 3f). Enzyme-Linked Immunosorbent Assay (ELISA) showed similar results with stronger human-complement binding for P9 and P57 versus Binder-47 (Supplementary Fig. 5a). These protein-protein interaction results are consistent with their enhanced in vitro hemolysis inhibition ability (Fig. 3b). Additionally, the P1–P60 group binders were screened against guinea-pig and rabbit sera, yielding 7 and 45 active binders, respectively (Supplementary Fig. 4b & Supplementary Data 1). At the same inhibitor concentration, the optimized P1–P60 group binders yielded more candidate binders that potently inhibited rabbit complement activity compared to guinea-pigs. In the future, optimizing partial diffusion using complement C9 structures from various species could yield better performance binders for more species.

## Crystal structure and the binding specificity of the C9 mini-protein inhibitors
To further validate the accuracy of our designs, recombinant mouse C9 was mixed with the P57 mini-protein inhibitor and subjected to crystallization trials. SEC and SDS-PAGE confirmed stable complex formation (Supplementary Fig. 6). However, no diffraction-quality crystals of the complex were obtained. To verify the accuracy of our

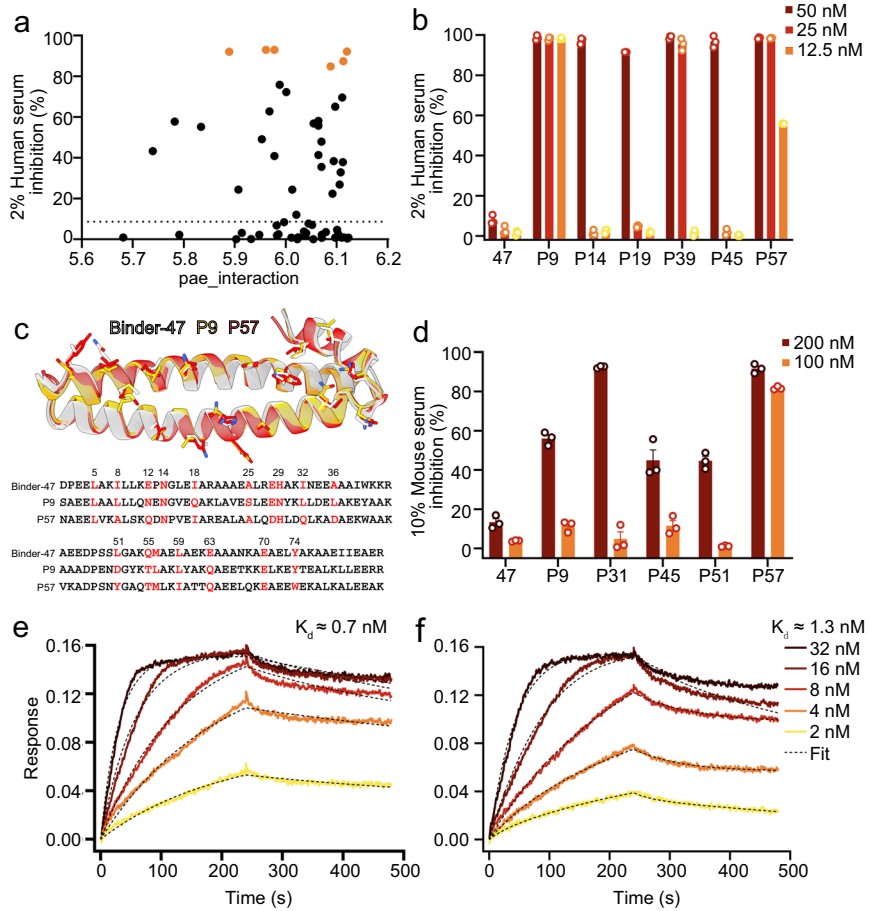

**Fig. 3 | Affinity maturation using partial diffusion. a** In vitro hemolysis inhibition rates plotted against pae_interaction scores for 60 partially diffusion optimized mini-protein inhibitors. Six binders exhibiting >80 % inhibition rate are highlighted in orange, the dashed line indicates the inhibition rate of the parental Binder-47. **b** Hemolysis inhibition was further profiled across a gradient of sequentially diluted binder concentrations. **c** Superimposition and sequence alignment of the parental Binder-47 (silver) and the partial diffusion optimized binders P9 (orange) and P57

(red). **d** The inhibition effect of six identified binders against 10% mouse serum. **e** BLI characterization of the interaction between mini-protein binder P9 and human C9. **f** BLI characterization of the interaction between mini-protein binder P57 and mouse C9. Graphs in **b, d** show mean ± SEM of $n = 3$ biologically independent samples. Source data are provided as a Source Data file (Predicted model for (**c**) is deposited to zenodo.org with accession code: 18530064).

designs, we started screening the crystals of the P9 and P57 binders alone. Initial binder crystal screening obtained protein crystals, but diffraction was poor. Two to three surface-exposed hydrophilic residues distant from the binding interface were mutated to alanine to enhance crystal diffraction[40–43] (Supplementary Table 1). Two P57 variants, P57-M4 and P57-M5, yielded high-quality crystals that diffracted to 1.4 Å and 1.8 Å, respectively (Supplementary Table 2). The Cα r.m.s.d of two crystal structures was 0.49 Å (Supplementary Fig. 7b) and superimposed onto the design model with Cα deviations of 0.85 Å (P57-M4) and 1.50 Å (P57-M5) (Fig. 4a & Supplementary Fig. 7a), demonstrating high consistency with the design model. Hemolysis inhibition assays showed that the crystal variants retained full inhibitory activity, indicating that surface alanine mutations do not impair P57 function (Fig. 4b). To further verify the interaction specificity between binders and C9, we performed C9 reconstitution experiments using C9-depleted human serum. C9-depleted human serum cannot cause hemolysis, reintroduction of purified human C9 to C9-depleted serum restored hemolytic activity, which was subsequently blocked by P9 and P57 (Fig. 4c), demonstrating that inhibition is strictly C9 dependent. Moreover, we performed BLI and ELISA assays using the well performed P9 and P57 mini-protein inhibitor against a panel of human complement proteins. BLI results showed that P9 specifically binds C9, but not C3, C4, C5, C6, C7, C8 complex (Fig. 4d). Since some complement proteins carry a polyhistidine tag, we performed direct

ELISA using C-terminal horseradish-peroxidase (HRP) labeled P57 (P57-HRP) (Supplementary Fig. 8). Similarly, ELISA data showed that P57-HRP interacts only with C9, not with other complement components (Fig. 4e). These results demonstrate that the mini-protein inhibitor is highly specific, binding specifically to C9. Finally, several key interface residues (positions 29, 36, 63) in P9 and P57 were mutated to arginine (Fig. 4a). Hemolysis inhibition assays revealed that, at a fixed binder concentration of 50 nM, mutations at these positions greatly attenuated the inhibition efficacy of both P9 and P57 against complement-mediated hemolysis (Fig. 4f). ELISA assays also showed that the binding of these mutants to human C9 was greatly reduced (Fig. 4g). These results indicate that P9 and P57 specifically inhibit soluble C9 to prevent the formation of MAC.

### Comparison of C9 mini-protein inhibitor with antibodies

Having identified two well behaved C9 binders, we next compared their inhibitory capacity with the clinically approved anti-C5 monoclonal antibody eculizumab and two commercialized C9 monoclonal antibodies X197 and E-3. X197 previously showed inhibitory activity in a non-standard hemolysis assay[33]. E-3 is a widely used anti-human C9 antibody. In our classical gradient hemolysis inhibition assay, P9, P57, and eculizumab exhibited IC50 values of 7.49, 11.67, and 4.79 nM, respectively (Fig. 5a), suggesting that C9 mini-protein inhibitors could achieve hemolytic inhibition similar to the C5 inhibitor. On the other

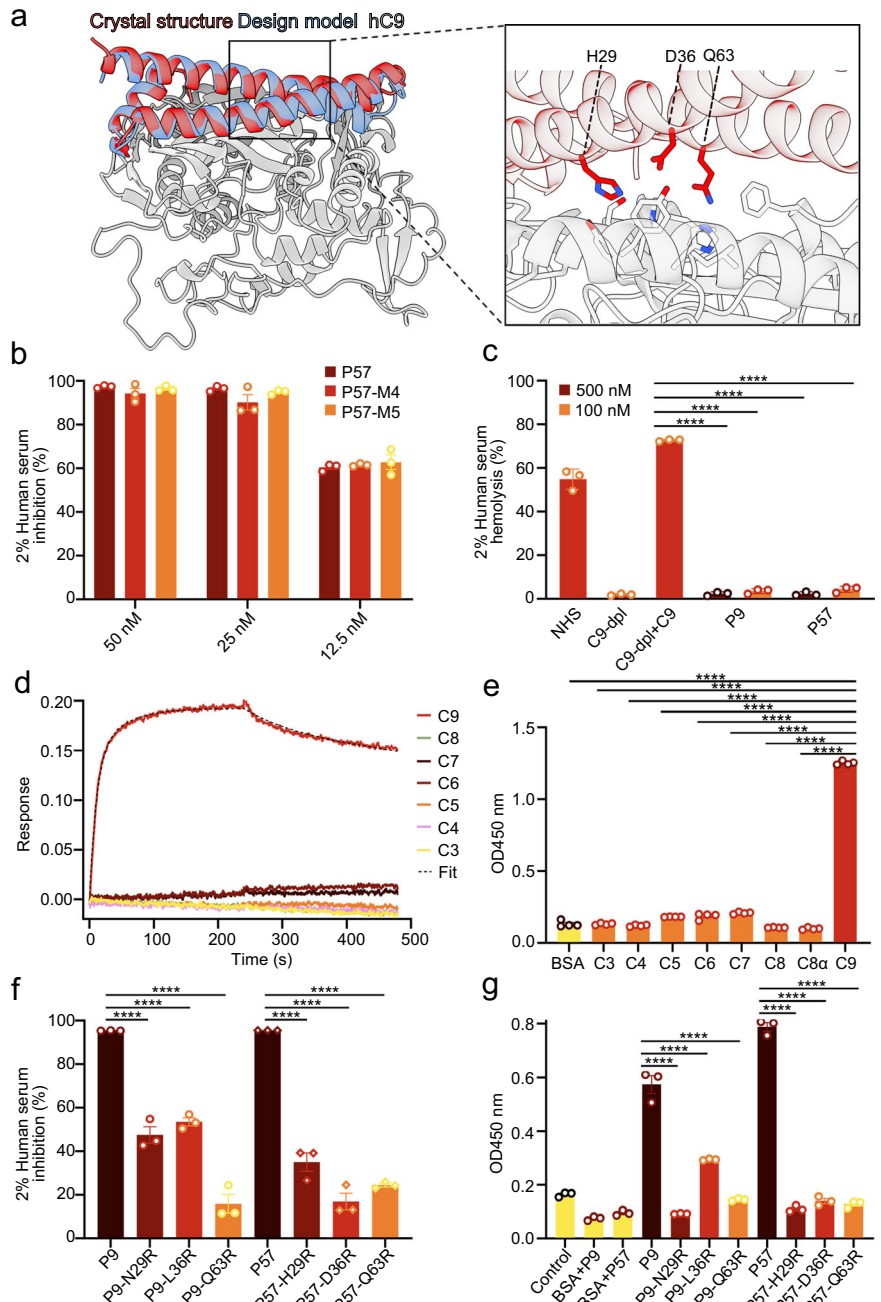

**Fig. 4 | Crystal structure and binding specificity of C9 mini-protein inhibitors.** **a** Superimposition of the P57-M4 crystal structure (red) with the designed model of P57 (blue) and human complement C9 (hC9, gray) complex. Zoom-in view shows three key interacting residues in P57. **b** Dose–responsive hemolysis inhibition assays for the crystal constructs P57-M4 and P57-M5. **c** In vitro complement C9 reconstitution assay using C9-depleted human serum and purified human C9 protein. **d** The BLI experiment detects the interaction between P9 and several complement components, including C3, C4, C5, C6, C7, C8, and C9. **e** Direct ELISA was employed to quantify the interaction between complement C3, C4, C5, C6, C7, C8 complex, C8α, C9, and P57-HRP. **f** In vitro hemolysis inhibition assay using mini-protein inhibitors with arginine mutations at key interaction sites. **g** Indirect ELISA was employed to quantify the interaction between complement C9 and mini-protein inhibitors, including corresponding site mutations to (**f**). Graphs in (**b, c, e, f** & **g**) show mean ± SEM from biologically independent samples ($n = 3$ for **b, c, f** & **g**; $n = 4$ for **e**). Statistical significances for (**b, c, e, f** & **g**) were determined by ordinary one-way ANOVA followed by two-sided Dunnett's multiple comparisons test. F values, degrees of freedom (df), and adjusted $P$ values are provided in the Supplementary Data 2. ****$P < 0.0001$. Source data are provided as a Source Data file (Predicted model for **a** is deposited to zenodo.org with accession code: 18530064).

hand, both C9 monoclonal antibodies were markedly weaker and failed to reach the efficacy of P9 or P57 (Fig. 5a). Against mouse serum, P57 displayed an IC50 of 93.80 nM, whereas eculizumab showed no detectable inhibition (Fig. 5b). We also established an ABO-incompatible hemolysis assay in vitro, both P9 and P57 effectively suppressed hemolysis (Fig. 5c). Given that the first C9 insertion is the kinetic bottleneck of MAC formation[25], to mimic inhibition of MAC

formation during acute hemolysis, we adjusted the classic hemolysis inhibition assay. Different from mixing serum with MAC inhibitors including eculizumab or binders before being added to red blood cell, human serum was added directly to red blood cells. After 2 min of reaction, the MAC inhibitors were added respectively, and the results indicated that the hemolysis inhibition rate of the C5 inhibitor was markedly lower than C9 mini-protein inhibitors (Fig. 5d). Additionally,

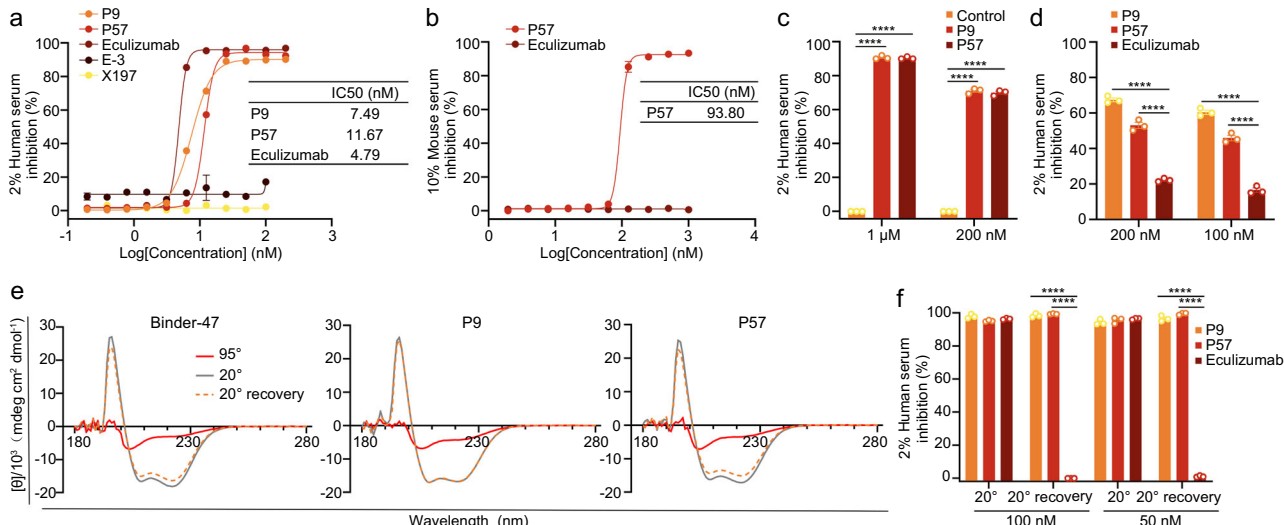

**Fig. 5 | Comparison of C9 mini-protein inhibitors with antibodies. a** In vitro gradient hemolysis inhibition assay mediated by mini-protein inhibitors, eculizumab, and anti-complement C9 antibodies (E-3, X197) in NHS. **b** In vitro gradient hemolysis inhibition assay mediated by P57 mini-protein binder. The IC50 values in (**a** & **b**) were calculated from the corresponding fitting curves. **c** The ABO-incompatible hemolysis assay was performed using the designed mini-protein inhibitors. **d** The adjusted in vitro hemolysis inhibition assay was performed, in which mini-protein inhibitors or Eculizumab were added after 2 min of complement activation. **e** The CD spectra confirmed the helical structure of the designed mini-protein binders (gray, 20 °C; red, 95 °C; orange dash, 20 °C followed by 95 °C). **f** An in vitro hemolysis inhibition assay was conducted to compare the biological activity of the designed mini-protein inhibitors before and after heat treatment, using Eculizumab as the reference standard. Graphs in (**a–d**, **f**) show mean ± SEM of *n* = 3 biologically independent samples. Statistical significances for (**c, d, f**) were determined by ordinary one-way ANOVA followed by two-sided Dunnett's multiple comparisons test. F values, degrees of freedom (df), and adjusted *P* values are provided in the Supplementary Data 2. ****$P < 0.0001$. Source data are provided as a Source Data file.

we tested the thermal stability and yield of the C9 mini-protein inhibitors, Circular dichroism (CD) data revealed that, Binder-47, P9 and P57 almost fully recovered their secondary structure after heating to 95 °C and cooling to 20 °C (Fig. 5e). Functionally, all three binders preserved their inhibitory activity following incubation at 95 °C for 5 min, whereas eculizumab lost activity under the same conditions (Fig. 5f). Mass spectrometry confirmed the intact molecular weights and retention of N-terminal methionine (Supplementary Fig. 9). Without optimization, shake-flask expression in 100 mL yielded 49.98 mg/L for P9 and 126.85 mg/L for P57 (Supplementary Fig. 10) and both binders exhibited excellent solubility.

**Inhibition of intravascular hemolysis in vivo**
Furthermore, we established intravascular hemolysis assays in mice using human serum[23]. Firstly, P9, P57, or eculizumab was pre-incubated with human serum prior to tail vein injection (Fig. 6a). The results showed that all three inhibitors could completely suppress free hemoglobin release without significant differences (Fig. 6b). Additionally, consistent with the in vitro experiments, the serum was injected without prior incubation with the inhibitors. At different time intervals, P9 or eculizumab was then administered via the tail vein (Fig. 6a). The C9 mini-protein inhibitor showed greater hemolytic inhibition than the C5 inhibitor eculizumab at all time points, and could almost completely inhibit hemolysis even after complement activation had started for 8 min, compared to eculizumab which began to fail at 6 min (Fig. 6c). These in vivo data demonstrated that C9 mini-protein inhibitors match eculizumab in potency under standard chronic conditions and provide a wider therapeutic window after acute complement activation.

## Discussion
To obtain high-affinity mini-protein inhibitors, we employed relatively large-scale cluster computing and generated approximately 25,000 binder scaffolds in the initial screening phase, which led to high computational costs. Iterative hotspot sampling inevitably increases

computational demand, yet it has been regarded as a necessary step. However, with the development of the next generation of binder design tools, such as BindCraft[44], the ability of automatic epitope identification and integrated backbone generation, sequence design, and structure prediction scoring may improve the success rate and reduce the costs required for computation. In the experimental screening phase, unlike the BLI-based protein-protein interaction screening, we directly used hemolysis inhibition experiments for functional screening. This approach enables more rapid and accurate identification of mini-protein inhibitiors with potent biological function. Indeed, in many biological processes, mini-protein inhibitors require high $K_{on}$ rates and lower $K_{off}$ rates to compete with natural ligands. Although a series of mini-protein inhibitors have been reported, there is significant variability in their properties. Some binders exhibited poor solubility and low expression levels, partly due to their small size and relatively hydrophobic binding interface, which poses obstacles to the further application of these mini-protein inhibitors. The C9 mini-protein inhibitors in this study exhibited lower hydrophobicity and better specificity at the binding interface, endowing the mini-protein inhibitors with better therapeutic prospects.

The complement C9 binding site targeted in this study has no reported natural ligands, and this region is highly conserved among mammals. In fact, many components of the ancient and conserved immune system, such as the complement system, can provide more design targets for de novo design, compensating for the shortcomings of traditional antibody screening methods and meeting the needs for disease treatment. Inhibiting complement C9 selectively can prevent MAC formation, allowing MAC to partially form but not ultimately destroy cells. Since complement C9 is located at the terminal end of the complement cascade, inhibiting C9 to some extent reduces the impact on the dynamic changes of other complement components. Theoretically, blocking C9 membrane insertion would weaken bactericidal and virucidal activity under normal conditions and thereby increase the risk of infection; indeed, individuals with C9 deficiency exhibit a markedly higher recurrence rate of invasive meningococcal disease[45,46]. In addition, inhibiting C9 insertion may slow immune-

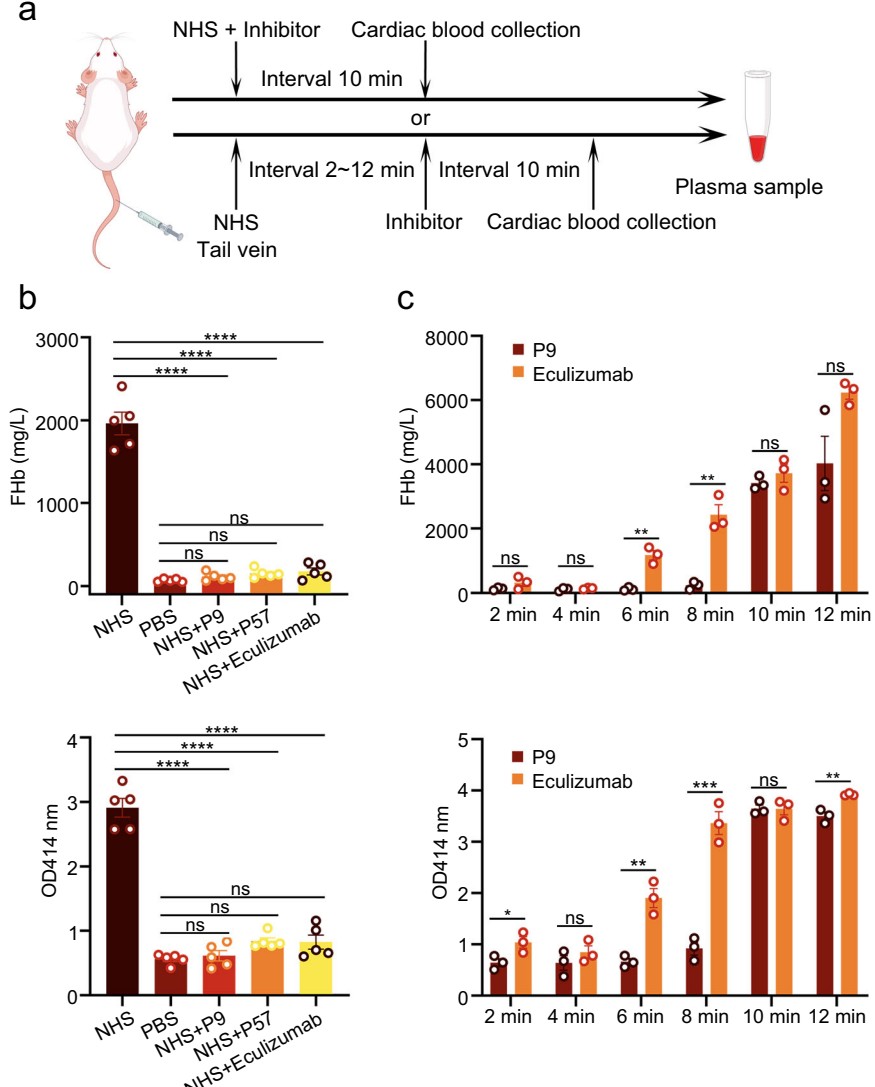

**Fig. 6 | Inhibition of intravascular hemolysis in vivo. a** Schematic representing the workflow for the in vivo intravascular hemolysis inhibition assay. **b** NHS was pre-incubated with the corresponding complement inhibitors and administered to mice via tail vein injection. The resulting intravascular hemolysis was quantified by measuring plasma free hemoglobin and the absorbance at 414 nm, allowing direct comparison of inhibitory activity among the de novo designed mini-proteins P9, P57, and Eculizumab. **c** NHS was injected via the tail vein into mice without prior incubation with corresponding complement inhibitors, inhibitors were injected via tail vein into mice after intervals of 2, 4, 6, 8, 10 or 12 min. The top panels of (**b**, **c**) show plasma free hemoglobin concentration; the bottom panels of (**b**, **c**) show quantified hemolysis by plasma absorbance at 414 nm. Graphs in (**b**, **c**) show mean ± SEM from biologically independent samples ($n = 5$ for **b**; $n = 3$ for **c**). Statistical significances were determined by ordinary one-way ANOVA followed by two-sided Dunnett's multiple comparisons test (for **b**), and two-sided unpaired t test (for **c**). F values, degrees of freedom (df), adjusted $P$ values, and exact $P$ values are provided in the Supplementary Data 2. ns, not significant, *$P < 0.05$, **$P < 0.01$, ***$P < 0.001$, and ****$P < 0.0001$. Some elements in **a** were created by Figdraw (www. figdraw.com). Source data are provided as a Source Data file.

complex clearance and exacerbate immune-complex–mediated nephritis[47]. Conversely, in chronic inflammatory disorders, preventing C9 insertion may offer therapeutic benefit against MAC-related pathologies such as age-related macular degeneration and other degenerative diseases[48,49]. However, the specific biological impact still needs further investigation. In fact, after the formation of MAC is inhibited, C3d induced extravascular hemolysis is a more urgent problem for patients with PNH[50]. Designing inhibitors targeting more upstream components, such as complement C3, is also challenging and meaningful[51]. The C9 mini-protein inhibitors demonstrated significant advantages in vivo acute hemolysis processes, which could be applied to diseases like AHTR and PCH providing a safeguard measure to inhibit hemolysis before substantial cell lysis occurs. For emergency treatment of acute hemolysis, extending the therapeutic window by just a few minutes is crucial for saving patients' lives and minimizing

tissue damage. Moreover, the small size of the C9 mini-protein inhibitors leads to a short half-life, allowing rapid drug clearance once the acute phase resolves. In the future, we hope to further evaluate the pharmacological mechanisms and efficacy of the C9 mini-protein inhibitors in the treatment of acute complement related diseases.

Considering future clinical applications, we need to conduct more detailed research and evaluation of the pharmacokinetics of the C9 mini-protein inhibitors. In diseases such as PNH, the long-term use of C9 mini-protein inhibitor necessitates a more thorough evaluation of their immunogenicity. The series of C9 mini-protein inhibitors reported in this study can serve as candidate scaffolds for immunogenicity testing, with sequence optimization to balance efficacy and immunogenicity. In addition, extending the half-life of mini-protein inhibitor through chemical modification, albumin binding, and other engineering strategies will be highly beneficial for improving compliance

with long-term treatment. In summary, further animal and clinical trials to evaluate the therapeutic effects of mini-proteins should be prioritized.

## Methods

### Ethics statement

This study was approved by the Ethics Review Board of Shandong Second Medical University. The animal experiments were conducted in strict adherence to the guidelines and regulations set forth by the Laboratory Monitoring Committee of Shandong Province, China, and were approved by the Experimental Animal Ethics Committee of Shandong Second Medical University (NO.2025SDL798). BALB/c mice were bred and raised under specific pathogen-free conditions, and were housed in a 12-h light/dark cycle environment and fed with standard feed. The temperature in the breeding room was maintained at 20–24 °C, and the relative humidity was kept between 50% and 60%. The studies involving human samples were approved by Medical Research Ethics Committee of Shandong Second Medical University (NO.2025YX196). The studies were conducted in accordance with the local legislation and institutional requirements. The participants provided their written informed consent to participate in this study.

### Computational binder design

The binder design workflow basically followed previously published protocols[35,36]. The initial target was the mouse C9 crystal structure (PDB: 6CXO)[52] and its sequence was re-numbered continuously. Hotspots were selected manually based on structural features. Generated backbones were relaxed with FastRelax and redesigned with ProteinMPNN, generating three sequences per scaffold. The ProteinMPNN scripts were lightly modified to incorporate multi-threading, ensuring high-throughput processing. All ProteinMPNN generated designs were predicted using the AlphaFold2 prediction protocol. For partial diffusion, an AlphaFold3 model of the Binder-47 in complex with human C9 was first generated. Partial diffusion was executed with diffuser.partial_T spanning 1–25 (step = 1), producing 1000 scaffolds per T value and further following similar ProteinMPNN and AlphaFold2 prediction protocols. AlphaFold3 predictions were performed using the online website (https://alphafoldserver.com/). The Supplement Figure 11 showed two view angles and the relevant scores with models presented in Figs. 1c, 2a, 3c, 4a, and Supplementary Fig. 7a.

### Protein expression and purification

Mini-protein inhibitor coding sequences were codon-optimized for expression in *E. coli*. Genes were synthesized and inserted between the NcoI and XhoI sites of pET-28a. For expression of human and mouse C9, mouse C9 (Q21-I548) and human C9 (Q22-K559) were codon-optimized for human cells, fused with an N-terminus mouse Ig kappa secretion signal (METDTLLLWVLLLWVPGSTGD) and a C-terminal (His)$_6$ or human IgG1 Fc tag, inserted between the XbaI and BamHI sites of the pcDNA3.4. Mini-protein inhibitor mutant genes were directly synthesized and constructed. All genes were synthesized and constructed by Universe Gene Technology or GENCEFE Biotech. For mini-protein inhibitor expression and purification, plasmids were transformed into *E. coli* BL21(DE3) and plated on LB agar plates overnight. A single colony was inoculated into 100 mL LB medium and grown at 37 °C. Expression was induced with 0.3 mM IPTG when OD600 reached 0.6–0.8, followed by 16 h culture at 24 °C and 220 rpm. Cells were harvested by centrifugation (10,000 × $g$, 5 min), resuspended in 25 mL PBS, lysed by sonication and clarified by centrifugation (15,000× $g$, 20 min). The supernatant was applied to 1.5 mL Ni-NTA resin (Qiagen), washed twice with 6× bead volumes of PBS containing 20–30 mM imidazole (pH 7.4), and eluted with PBS plus 300 mM imidazole (pH 7.4). Eluted proteins were analyzed on 4–20% SDS–PAGE (GenScript) and visualized with Coomassie Brilliant Blue. For complement C9 expression, plasmids were sterile-filtered and transiently expressed by Novoprotein using their mammalian expression system. After 5 days expression, supernatants were harvested. (His)$_6$-tagged C9 was purified using Ni Smart beads (Smart-Lifesciences); Fc-tagged C9 was captured with rProtein A/G beads (Smart-Lifesciences), following the manufacturer's protocols. Affinity-purified C9 was further purified by Size-exclusion chromatography (SEC). For C9 mini-protein complex formation, C9 and mini-protein were mixed at a 1:5 molar ratio, incubated at room temperature for 30 min, and subjected to SEC on a Superdex 75 Increase 10/300 GL column (GE Healthcare) using an FPLC system (Union-biotech, UEV25D). Protein concentrations were determined by measuring A280 on a multimode microplate reader (Tecan Spark).

### In vitro hemolysis inhibition assay

For in vitro hemolysis inhibition assay, designed binders or antibodies of different concentrations were pre-incubated with mouse serum in a 96-well plate at 4 °C for 45 min, antibody-sensitized sheep erythrocytes and GVB$^{++}$ buffer (10 mM barbital, 145 mM NaCl, 0.15 mM CaCl$_2$, 0.5 mM MgCl$_2$, and 0.1% gelatin) were then added, and the mixture was incubated at 37 °C for 30 min. Rabbit anti-sheep erythrocyte antibody (Beijing Bersee Biotechnology, BM351Y) was used to sensitize sheep erythrocytes (Zhengzhou Pingrui Biotechnology, ZPRMYHXB-100/4) at a dilution of 1:400 when using mouse serum. The blank control group was treated with an equal volume of GVB$^{++}$ buffer. The positive control group obtained packed red blood cells by centrifuging the red blood cell suspension, and used an equal volume of ddH$_2$O to completely lyse the red blood cells. The mouse serum concentration used in Fig. 2c was 15%, all other mouse serum concentrations were 10%, and the normal human serum (NHS) concentration was 2%. After incubation, samples were centrifuged to pellet the remaining intact erythrocytes, and supernatants were collected. Hemoglobin release from erythrocyte lysis was detected by measuring the optical density of the supernatant at 414 nm using a multimode microplate reader (Tecan Spark). The sera of different species, including BALB/c mouse, guinea pig and rabbit, were all collected from 8-week-old male animals. For serum collection, a total of 50 BALB/c mice, 1 guinea pig, and 1 rabbit were used. All this animials were purchased from Shandong Pengyue Laboratory Animal Technology. Antibodies were purchased accordingly: eculizumab (AstraZeneca), C9 antibody (E-3; Santa Cruz Biotechnology, sc-390000), X197 antibody (Hycultbiotech, HM2111).

A complement C9 reconstitution hemolysis inhibition assay shown in Fig. 4c was performed using 2% C9-depleted human serum (Complement Technology, A326) supplemented with 600 ng/well purified human C9 protein (Complement Technology, A126). For ABO-incompatible hemolysis assay, human red blood cells from blood group B$^+$ donors, 1% were exposed to 50% incompatible human serum from blood group A$^+$ donors in GVB$^{++}$ buffer. For the adjusted acute hemolysis assay, human serum was used to induce hemolysis in sheep erythrocytes at 37 °C for 2 min, followed by the addition of either eculizumab or designed binders, and the reaction was continued at 37 °C for an additional 10 min. Samples were handled and detected as described in the above hemolysis assay. Ten healthy male volunteers aged 20–35 years, with no history of infectious or autoimmune diseases, were recruited for blood collection. After collection, the blood was first left to stand at room temperature for 30 min, and then centrifuged at 3000 g at 4 °C for 10 min to obtain serum. For the hemolysis inhibition assay, animal blood was obtained by cardiac blood sampling, and serum was extracted through a similar process. To exclude the influence of sex on complement activity, only blood from male humans or animals was collected. Percentages of hemolysis = $[(OD_{sample} - OD_{blank})/(OD_{positive} - OD_{blank})] \times 100$; Percentage of inhibition = $[1 - (OD_{sample} - OD_{blank})/(OD_{positive} - OD_{blank})] \times 100$.

**ELISA.** Indirect ELISA was employed to detect interaction between native or mutant binders and soluble complement C9. Microtiter plates (Corning) were coated overnight with human C9 (Complement

Technology, A126) at 0.5 µg per well. After discarding the coating solution, plates were washed three times with PBST (PBS with 0.05% Tween-20, pH 7.4). Wells were blocked with 5 % BSA at 37 °C for 2 h. Mini-protein inhibitors were added sequentially and incubated at 37 °C for 2 h, followed by three PBST washes. A Rabbit anti-(His)$_6$ HRP-conjugated antibody (Proteintech, HRP-84814) was applied at 1:5000 dilution and incubated at 37 °C for 1 h. After washing, TMB (Thermo Fisher) was used as chromogenic substrate, and the reaction was stopped with 2 N $H_2SO_4$. Absorbance was measured at 450 nm using a multimode microplate reader (Tecan Spark).

Direct ELISA was used to assess interaction between P57-HRP and multiple complement components including complement C3 (Complement Technology, A113), C4 (Complement Technology, A105), C5 (Complement Technology, A120), C6 (Sino Biological, 12426-H08H), C7 (Sino Biological, 13848-H08H), C8 complex (Complement Technology), C8α (Abcam, ab316549) and C9 (Complement Technology, A126). An extra cysteine was introduced before the C-terminal stop codon in the P57 expression plasmid to express the P57-C protein and enable HRP labeling. After expression and purification, P57-C was labeled with the HRP Maleimide Activator Labeling Kit (TCI Chemicals). Briefly, HRP and P57-C were mixed with the molar ratio of 1:5 with the reaction volume of 500 µL in PBS and incubated overnight. Free HRP was removed by Ni-NTA based affinity chromatography, and unlabeled P57-C was separated by size-exclusion chromatography, yielding pure P57-HRP (Supplementary Fig. 8).

**BLI.** The BLI experiments were conducted using an OCTET RED96E system (ForteBio) and the results were processed with the integrated software (Fortebio data analysis 12.0.1.2). For mini-protein binder gradient binding assay, Fc-tagged human or mouse C9 proteins were diluted to a final concentration of 5 µg/mL and loaded onto the Protein A sensor (ForteBio), with a loading time of 600 s. For multiple complement components binding assays, C3, C4, C5, C6, C7, C8 and C9 were biotinylated using chemical methods. Complement protein (100 µl, ~0.5 mg/mL in PBS) was biotinylated using a commercial kit (Genemore) following the manufacturer's protocol. Briefly, the protein was incubated with a predetermined molar excess of the biotinylation reagent for one hour at room temperature. The reaction mixture was purified using a desalting column to remove unconjugated biotin. Biotinylated complement proteins were diluted to a final concentration of 5 µg/mL and loaded onto the SA sensor (ForteBio), with a loading time of 600 seconds. Mini-protein inhibitors were serially diluted to achieve a range of concentrations from 128 to 2 nM, and the assay buffer was PBST. The concentration for P9 was 32 nM in multiple complement components binding assays (Fig. 4d). Except for the Binder-47 binding to human complement C9 in Fig. 2f, where the association times were 180 s, the association and dissociation times for other experiments were all set to 240 s. The flow rate was 600 rpm for loading and 1000 rpm for other steps, all experiments were conducted at 25 °C. Global fitting was applied for Figs. 2f, 3e, 3f and Supplementary Fig. 5b using a 1:1 fitting model. BLI analysis of mini-protein Binder-47 binding to mouse complement C9 yielded $R_{max}$ of 1.1505, $R^2$ of 0.9821, $X^2$ of 0.0052, and $K_d$ of 3.3 nM; binding to human complement C9 yielded $R_{max}$ of 0.3737, $R^2$ of 0.997281, $X^2$ of 0.000364, and $K_d$ of 22 nM (Fig. 2f). Mini-protein P9 with human complement C9 yielded $R_{max}$ of 0.1881, $R^2$ of 0.976, $X^2$ of 0.0118, and $K_d$ of 0.7 nM (Fig. 3e) and P57 with mouse complement C9 yielded $R_{max}$ of 0.2211, $R^2$ of 0.9713, $X^2$ of 0.011, and $K_d$ of 1.3 nM (Fig. 3f). P57 with human complement C9 resulted in Rmax of 0.1388, $R^2$ of 0.9724, $X^2$ of 0.0064, and $K_d$ of 2.1 nM (Supplementary Fig. 5b). For Fig. 4d, local fitting was applied using a 1:1 fitting model with Rmax of 0.1917, $R^2$ of 0.9661, $X^2$ of 0.0103. To subtract the background, each independent experiment included a reference well without mini-protein inhibitor added in association step. The buffer used for sensor activation, baseline stabilization, and protein dilution was PBST.

**CD.** Circular dichroism (CD) characterization was conducted using a Chirascan V100 instrument (Applied Photophysics). Briefly, 120 µL of 0.04 mM mini-protein inhibitor in PBS was placed into a quartz cuvette. Sequential wavelength scans were performed at 20 °C, followed by heating to 95 °C, and then a return to 20 °C to observe recovery after heating. Far-ultraviolet CD spectra were recorded within the 180 - 280 nm range. The average protein spectrum was obtained by subtracting the buffer baseline spectrum and this measurement was replicated three times for accuracy. The CD signal was converted to mean residue ellipticity ([θ]), dividing the corrected spectra by the product of the number of amino acid residues (N), the protein concentration (C), and the cuvette path length (L, which is 1 mm).

### Crystallization and structural determination
Initial crystal screening was conducted using kits purchased from Hampton Research and Rigaku, which included Crystal Screen, PEGRx, and Wizard Classic 1-4. Initial screenings, along with subsequent optimizations, were carried out using a two-position deck mosquito LCP (SPT Labtech) within 96-well crystallization plates using the sitting-drop method. The crystals of the initially screened P9 and P57 binders had poor diffraction quality. To improve the crystal quality, 2-3 surface residue mutations were introduced into mini-protein inhibitors. Sequences are shown in Supplementary Table 1. (His)$_6$ tag and design sequences were directly fused without GGS linker. The crystallization conditions for P57-form 1 were 0.1 M Citric acid pH 3.5, 34% v/v Polyethylene glycol 200. For P57-form 2 were 0.2 M Magnesium chloride hexahydrate, 0.1 M Tris pH 7.0 and 10% w/v Polyethylene glycol 8,000. The crystals were subjected to diffraction and indexing using an in-house X-ray diffraction system (Rigaku) utilizing Cu Kα radiation with a wavelength of 1.54 Å. The structure was solved using the molecular replacement method, employing the designed models as search templates. The P57-form 2 was first solved by using P57 as the search model. The P57-form 1 structure was solved by using the crystal structure of P57-form 2 as the search model. Molecular replacement and structure refinement were executed using the PHENIX suite[53] and COOT[54]. Visualization of the structures was performed using ChimeraX[55] and PyMOL (The PyMOL Molecular Graphics System, 2002). The collection and refinement statistics are comprehensively presented in Supplementary Table 2.

### In vivo intravascular hemolytic studies
To establish a complement-driven intravascular hemolysis assay, 8-week-old male BALB/c mice received NHS via the tail vein. Two protocols were employed: serum and inhibitors were pre-mixed and administered at a single 200 µl tail vein injection, doses were calculated based on mouse body weight: NHS 1.5 µl/g, eculizumab 6 µg/g, and anti-C9 mini-protein inhibitors (P9, P57) 6 µg/g, in sterile PBS. Negative controls received equal volume of PBS. For the classic intravascular hemolysis assays, serum was incubated with inhibitors (or PBS) and tail vein injected immediately. Mice were euthanized 10 min after injection with inhibitors. Whole blood was collected by cardiac puncture, and plasma was separated via centrifugation at 3000 × g, 4 °C, 10 min. The Percentage of hemolysis was quantified by measuring the absorbance at 414 nm, free hemoglobin was detected using a kit (Thermo Fisher) following manufacturer's instructions.

For the adjusted intravascular hemolysis assays, 100 µl NHS was injected via the tail vein and allowed to circulate for defined intervals (2, 4, 6, 8, 10, or 12 min); inhibitors (100 µl) were then delivered at the same dose as above. Mice were euthanized 10 min after injection with inhibitors. NHS, inhibitor, and blood samples were processed as described above.

### Statistics
Data in the figures were presented as mean ± SEM. Statistical analyses were performed using one-way analysis of variance for multiple-group

comparisons. An unpaired, two-sided *t* test was used for comparisons between two groups. Statistically significant *P* values were summarized in Supplementary Data 2. The gradient hemolysis assay used a standard four-parameter dose–response inhibition function to calculate IC50 (50% inhibition) values. All analyses were carried out using GraphPad Prism 8 for Windows (GraphPad Software, www.graphpad.com).

### Reporting summary
Further information on research design is available in the Nature Portfolio Reporting Summary linked to this article.

## Data availability
Coordinates and structure files have been deposited to the Protein Data Bank with accession codes 9X1W (P57-M4), 9X1X (P57-M5). Full raw data including all tested models are available on zenodo.org (accession code: 18530064). The final sequences of the designed inhibitors optimized by partial diffusion are provided in Supplementary Data 1. Source data is available with this paper as a Source Data file. Source data are provided with this paper.

## Code availability
The code used to generate the scaffolds of mini-protein binders in this study has been previously published[35] and is publicly available on GitHub at https://github.com/RosettaCommons/RFdiffusion, under BSD 3-Clause License. The code used to sequence design and complex structure generation has been previously published[56] and is publicly available on GitHub at https://github.com/nrbennet/dl_binder_design, under MIT License (for the main codebase) and Apache-2.0 License (for AlphaFold2 module). The code we developed has been uploaded to Zenodo with https://doi.org/10.5281/zenodo.18530064, under BSD 3-Clause License.

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

## Acknowledgements

This work was supported by grants from the National Natural Science Foundation of China (32501303 to B.Y., 82303251 to N.W., 81873883 to S.L., 82000525 to M.F.L.), the Shandong Provincial Natural Science Foundation, China (ZR2022QC209 to B.Y., ZR2023QH202 to N.W.). This work was also supported by the Shandong Provincial Science and Technology Support Plan for Youth Innovation in Universities (10438202502 to B.Y.) and the Graduate Student Research Grant from Shandong Second Medical University (2025YJSCX022). We would like to thank the Protein Characterization and Crystallography Facility of Westlake University for help in sample analysis; the Mass Spectrometry & Metabolomics Core Facility of Westlake University for sample analysis; and the Westlake University HPC Center for computation assistance.

## Author contributions

B.Y. and S.L. designed the research. B.Y. made the designs. M.L., N.W., X.F., G.W., and Z.Z. performed most of the experiments with the help of Y.Y., T.X., Y.Z., J.P., D.W., M.F.L., Y.L., J.T., Z.J. All authors analyzed data. B.Y., S.L,. and L.C. supervised research. B.Y., S.L., and M.L. wrote the manuscript with the input from the other authors. All authors revised the manuscript.

## Competing interests

B.Y., S.L., M.L., X.F., and N.W. are coinventors on a patent application for invention (applicant: Shandong Second Medical University; application number: CN202511704580.7; status: under substantive examination) that incorporates the usage of mini-protein binders P9 and P57 in hemolysis inhibition. The remaining authors declare no competing interests.
