## [Transparent Peer Review file · Nature Communications]

De novo design of miniprotein inhibitors targeting complement C9 to block membrane attack complex assembly

Corresponding Author: Professor Bowen Yu

Version 0:

Reviewer comments:

Reviewer #1

(Remarks to the Author)

This is an exciting study that reports the development of a novel protein designed to bind C9. The protein was designed using the murine structure as a template, and using ML-approaches made a series of potential binders by optimizing interactions across 3 specified hot-spots.

Minor comment: Figure 1a. Please adjust the color of C7, it is hard to see the distinction with C5 in some views.

They next leverage AF3 to prioritize binders and test functionality in hemolysis assays (both human and murine serum) and binding assays (BLI). They show that key residues of the predicted interface are conserved across human/mouse and other mammals.

Minor comment: Figure 2a. Needs to be clear in the legend this is an AF3 prediction, not an actual structure.

The authors then use partial diffusion to mature their designs and test for improved binding and functionality.

They then validate the designed C9 mini-binder with CD and by solving the crystal structure of 2 designs. They validate interaction interfaces via ala mutagenesis and testing the functionality in a red blood cell lysis assay.

My main concern is regarding specificity. The authors clearly show that the newly designed protein can bind recombinant C9 in vitro assays, but they need to show that it doesn't bind other complement proteins that have a very similar MACPF domain, for example C6, C7 or C8. In particular, the MACPF domain of C8a is very similar to C9.

Finally they show that in a mouse model the mini-c9 binding protein has similar if not better potency than clinically used terminal pathway inhibitors.

The structural biology aspects of the manuscript are sound and the statistics for the refined model are good.

Other minor corrections:

"Given that C9 insertion is the kinetic bottleneck of MACformation",

Be careful, this paper showed that the 1st C9 addition is the kinetic bottleneck rather than polymerization of more C9s into 1 mac, it didn't give rates for C7, or C8 addition.

"thereafter about 20 soluble C9 molecules rapidly oligomerize to complete the pore".

It should be 18 copies of C9 (doi: 10.1038/ncomms10587)

“E.coli” should be in italics throughout

Reviewer #2

(Remarks to the Author)

This paper describes high affinity mini-protein inhibitor targeting complement C9. Although AI-based technology for generating high-affinity mini-protein binders is not novel, it has not yet been applied for designing inhibitors of complement. Since the identified molecule targets a binding site in C9 that has no reported natural ligands, the work is significant for the complement field.

Major points:

1. The Methodology description is very thorough (especially the parts on computational design, structure analyses). I see this as a positive point, but also raises the questions whether the paper is more suited for a Methodology oriented journal (Nature Methods)
2. The specificity of the mini-proteins for complement C9 is not accurately assessed. In Fig. 4C, the authors claim to demonstrate specificity since the molecule only blocks lysis in the presence of C9. The interpretation is incorrect. Indeed there are only functional MAC pores when C9 is present, but the inhibitor can still act on other complement components. Especially binding to proteins homologous to C9 (C6, C7, C8) should be directly assessed by testing binding to purified complement proteins (for example ELISA). Other non-homologues (C3, C4, C5) should be included as control. Also in BLI, specificity controls should be included!!

Minor points:

- A supplemental figure /schematic of binder design/selection process would be helpful.
- Add the serum percentage to the relevant figures. This is important to interpret the results and to understand which pathways may be contributing to response.

[Introduction]

- o C5 cleavage is the initiating event for MAC formation. Add to text.
- o Reference 15 states that there are 18 copies of C9 (as well as other literature). Why do you state “about 20?”
- o Mention CD59, which binds C9 in the extended state. Makes a stronger case for needing a binder that prevents the conversion from closed to extended.

[Results]

- o In the screening of the initial set of binders, they mention effect of the classical pathway but then use 10%/15% serum. At this concentration AP should also be active.
- o Fig 1. Better description of the criteria used to choose the potential binder sites.
- o During the binder generation, percentages are noted. Add a sentence explain what values are good.
- o Jumping between AF2/AF3. Provide an explanation for switching between the two versions.
- o Fig 1C not mentioned in text.
- o Fig 2A not mentioned in before B-D text.
- o Fig 2E Specify human/mouse in the 2E.
- o Suppl 3 – nothing was mentioned about the potent effect of the inhibitors in rabbit serum.
- Figure 4d – N29R does cause 50% inhibition whereas no binding is detected at all (fig. 4e). What would be the explanation for this?
- Fig 5E – adjust the x-axis to better read the scale
- In the last results section, the final sentence is more appropriate for the discussion. “For emergency treatment of acute hemolysis, extending the therapeutic window by just a few minutes is crucial for saving patients’ lives and minimizing tissue damage.
- 5D. why this approach of first allowing lysis and then add inhibitors. Repeat in the ‘classical way’

[Discussion]

- For discussion, consider expanding to include the following
- o Computation resources required to pursue such a study. It’s only briefly mentioned and much of the emerging software is also computationally expensive.
 - o Impact of depleting C9 on native state, chronic conditions, infection, etc.
 - o Kon rates are discussed but what about Koff ?
 - o Longterm treatment of binder?

Reviewer #3

(Remarks to the Author)

Li et. al. present their design of mini-protein binders for complement C9, binders capable of inhibiting MAC formation. Using recently established protein design methods to find C9 binders (RFDiffusion, ProtMPNN, AlphaFold2) they discover a number of binders to monomeric C9, some with impressively high affinity, and solve the structure of a bound complex, which is very similar to the predicted design. The in vitro tests reveal that these new inhibitors have potent activity, and in many respects outform an approved antibody biologic. This is an experimentally solid, well-presented study that is an excellent addition to the growing number of studies that show the power diffusion methods in de novo protein design. Their argument about the value of early experimental analysis, and some limits of computational metrics, will be valuable to the field. And the new biologics found have clear biomedical potential.

I would recommend accept, I only have minor changes, mostly the clarify some statements:

“and the inhibitors are kinetically disadvantaged.” What are “the” inhibitors here? Existing monoclonal antibodies? Please clarify.

“ Despite this therapeutic rationale, C9-directed inhibitors remain rarely reported.” If they have been reported at all, references are needed.

“Moreover, the relatively high serum concentration of C9 requires that C9 inhibitors have high affinity” Is this right, I would expect the opposite. If the concentration of target is high, then any binder with a K_d below this concentration will bind. High doses might be required, but not high affinity.

“we can now generate binding proteins for any structural epitope” this could be too strong a statement. Indeed, in the same paragraphy it is claimed that “C9 remains challenging”

“ The soluble, monomeric crystal structure of mouse C9” Why not the human, or the predicted human structure (that is used later)?

For the “sheep red blood cell hemolysis inhibition assay” the organism of the serum should be stated in the main text. As written, it could be mistaken for sheep serum.

“In the future, optimizing partial diffusion using complement C9 structures from different species will result in better performance binders.” What does “better” mean in this context? Able to bind more species, or bind human better?

“were mutated to alanine to enhance diffraction”. Not obvious to me why this would work, and no reference given.

Version 1:

Reviewer comments:

Reviewer #1

(Remarks to the Author)

all comments and issues have been addressed by the authors.

Reviewer #2

(Remarks to the Author)

The authors have addressed all concerns in an appropriate fashion

Reviewer #3

(Remarks to the Author)

All requested revisions are well addressed. I recommend publish, its an excellent study of its type.

Point-by-point response to reviewers' comments on "De novo design of high affinity mini-protein inhibitor targeting complement C9 to block the final assembly step of membrane attack complexes"

Reviewer #1:

This is an exciting study that reports the development of a novel protein designed to bind C9. The protein was designed using the murine structure as a template, and using ML-approaches made a series of potential binders by optimizing interactions across 3 specified hot-spots:

(1) Minor comment: Figure 1a. Please adjust the color of C7, it is hard to see the distinction with C5 in some views.

Thanks, we change the color of C7 to yellow to better distinguish it from C5 in **Figure 1a (p. 25)**.

(2) Minor comment: Figure 2a. Needs to be clear in the legend this is an AF3 prediction, not an actual structure.

We have provided a clear explanation in the figure legend (**Fig. 2a**) (**p. 26, line 757-760**).

(3) The authors then use partial diffusion to mature their designs and test for improved binding and functionality. They then validate the designed C9 mini-binder with CD and by solving the crystal structure of 2 designs. They validate interaction interfaces via ala mutagenesis and testing the functionality in a red blood cell lysis assay. My main concern is regarding specificity. The authors clearly show that the newly designed protein can bind recombinant C9 in vitro assays, but they need to show that it doesn't bind other complement proteins that have a very similar MACPF domain, for example C6, C7 or C8. In particular, the MACPF domain of C8a is very similar to C9.

Thanks, this is indeed a critical issue. To determine whether the mini-protein inhibitor interacts with any other complement components, we performed ELISA and BLI assays using the well performed P9 and P57 mini-protein inhibitor against a panel of human complement proteins including C3, C4, C5, C6, C7, C8 complex, C8 α , and C9. The results demonstrate that the mini-protein inhibitor is highly specific, binding specifically to C9 but not other complement components (**Response Fig. 1**) (**p. 7-8, line 225-233**). We added the **Response Figure 1 Figure 4d & 4e**. The specific experimental details are described in the **materials and methods** section (**p. 15-16, line 435-460**).

Response Figure 1. (a) The BLI experiment detects the interaction between P9 and complement components, including C3, C4, C5, C6, C7, C8, and C9. **(b)** Direct ELISA was employed to quantify the interaction between complement C3, C4, C5, C6, C7, C8 complex, C8α, C9 and P57-HRP.

(4) Other minor corrections:

“Given that C9 insertion is the kinetic bottleneck of MAC formation”. Be careful, this paper showed that the 1st C9 addition is the kinetic bottleneck rather than polymerization of more C9s into 1 mac, it didn’t give rates for C7, or C8 addition.

“thereafter about 20 soluble C9 molecules rapidly oligomerize to complete the pore”.It should be 18 copies of C9 (doi: 10.1038/ncomms10587)

“E.coli” should be in italics throughout.

Thanks, we notice that these statements are indeed incorrect, and we have made corresponding modifications (p. 8, line 253-254), and 18 copies of C9 has been changed correctly (p. 2, line 68-70).

E.coli has been changed correctly.

Reviewer #2:

This paper describes high affinity mini-protein inhibitor targeting complement C9. Although AI-based technology for generating high-affinity mini-protein binders is not novel, it has not yet been applied for designing inhibitors of complement. Since the identified molecule targets a binding site in C9 that has no reported natural ligands, the work is significant for the complement field:

(1) The Methodology description is very thorough (especially the parts on computational design, structure analyses). I see this as a positive point, but also raises the questions whether the paper is more suited for a Methodology oriented journal (Nature Methods).

We sincerely appreciate your positive appraisal. Since our study primarily integrates previously published methods rather than introducing novel methodological advances, we considered *Nature Communications* to be the most suitable journal. We truly appreciate your encouragement.

(2) The specificity of the mini-proteins for complement C9 is not accurately assessed. In Fig. 4C, the authors claim to demonstrate specificity since the molecule only blocks lysis in the presence of C9. The interpretation is incorrect. Indeed there are only functional MAC pores when C9 is present, but the inhibitor can still act on other complement components. Especially binding to proteins homologous to C9 (C6, C7, C8) should be directly assessed by testing binding to purified complement proteins (for example ELISA). Other non-homologues (C3, C4, C5) should be included as control. Also in BLI, specificity controls should be included!!

Thanks, this is indeed a critical issue, also raised by Reviewer #1. To determine whether the mini-protein inhibitor interacts with any other complement components, we performed ELISA and BLI assays using the well performed P9 and P57 mini-protein inhibitor against a panel of human complement proteins including C3, C4, C5, C6, C7, C8 complex, C8 α , and C9. The results demonstrate that the mini-protein inhibitor is highly specific, binding specifically to C9 but not other complement components (**Response Fig. 1**) (p. 7-8, line 225-233). We added the **Response Figure 1 to Figure 4d & 4e**. The specific experimental details are described in the **materials and methods** section (p. 15-16, line 435-460).

(3) A supplemental figure /schematic of binder design/selection process would be helpful.

Thanks, we have added the **Supplementary Fig. 1** to illustrate our binder design and selection process (**Response Fig. 2**).

Response Figure 2. Schematic representation of the complement C9 binder design and selection process. The initial design and affinity maturation of the C9 mini-protein binder followed a workflow of hotspot selection, scaffold generation, sequence design, structure prediction, scoring, and screening. Detailed information is provided in the **methods**.

(4) Add the serum percentage to the relevant figures. This is important to interpret the results and to understand which pathways may be contributing to response.

Thanks, the detailed serum concentration used was added on the y-axis of each hemolysis-inhibition assays (Fig. 2 & 3 & 4 & 5).

(5) [Introduction]

- C5 cleavage is the initiating event for MAC formation. Add to text.*

Thanks, we have added this sentence to the introduction and provide relevant descriptions (p. 1-2, line 53-57).

- Reference 15 states that there are 18 copies of C9 (as well as other literature). Why do you state “about 20?”*

Thanks, we noticed this misstatement and made the corresponding correction (p. 2, line 68-70).

- *Mention CD59, which binds C9 in the extended state. Makes a stronger case for needing a binder that prevents the conversion from closed to extended.*

Thanks, we have added relevant descriptions about the working mechanism and the inspiration of CD59 in the introduction section (**p. 2, line 70-80**).

(6) [Results]

- *In the screening of the initial set of binders, they mention effect of the classical pathway but then use 10%/15% serum. At this concentration AP should also be active.*

Thanks, we have corrected the relevant statements (**p. 5, line 145-147**).

- *Fig 1. Better description of the criteria used to choose the potential binder sites.*

Thanks, we have added the relevant statements (**p. 4, line 112 -114**).

Typically, binder design favour surface-exposed hydrophobic residues as hotspots. We carefully checked the C9 interaction region for spatially clustered, interface-critical hydrophobic residues and grouped them into several sets. In addition, our empirical experience indicated that exposed β -sheet region such as Site-1 yielded higher success rate. And we made a **Supplement Figure 1** to introduce our screening and optimization process.

- *During the binder generation, percentages are noted. Add a sentence explain what values are good.*

Thanks, we have added relevant descriptions in the main text (**p. 4, line 118-125**).

In the scoring scheme, ‘pae_interaction’ is the predicted aligned error that AlphaFold2 assigns to the binder–target interface. It is the key metric for assessing how confident the model is in the interface prediction: the lower the value, the more reliable the prediction and the better the spatial fit between binder and target. Designs with pae_interaction < 10 are generally considered acceptable. Complementarily, ‘plddt_binder’ measures the per-residue confidence of the binder alone. A high value indicates that the sequence is likely to fold into a stable, well-defined monomeric structure; a threshold of > 90 is commonly required^{1,2}.

- *Jumping between AF2/AF3. Provide an explanation for switching between the two versions.*

Thanks, we have added several sentences in the main text to explain why AlphaFold3 was employed (**p. 5, line 158-160**).

After large-scale MPNN sequence design, all candidates were scored with the early-released AlphaFold2-complex prediction pipeline that has been widely adopted in binder design. Owing to the lack of a crystal structure for soluble monomeric human complement C9 and the potentially higher accuracy of AlphaFold3 in modeling protein–protein complexes, we employed AlphaFold3 to predict the single complex between human C9 and Binder-47 and subsequently used this model as the initial template for partial-diffusion calculations.

- *Fig 1C not mentioned in text.*

Thanks, we have added the citation for **Figure 1c** (p. 4, line 136-138).

- *Fig 2A not mentioned in before B-D text.*

Thanks, we have added the citation for **Figure 2a** before **B-D** (p. 5, line 142-144).

- *Fig 2E Specify human/mouse in the 2E.*

Thanks, we have included species labels in **Figure 2e** (p. 26).

- *Suppl 3 – nothing was mentioned about the potent effect of the inhibitors in rabbit serum.*

Thanks, we have added a description of the potent inhibition effect of rabbit complement activity in the main text (p. 6-7, line 198-204).

- *Figure 4d – N29R does cause 50% inhibition whereas no binding is detected at all (fig. 4e). What would be the explanation for this?*

Thank you for your question. Indeed, the results of the hemolysis assay and the ELISA experiment do not fully align. Overall, mutations at these sites greatly weaken the binding ability of P9 and P57. Since the hemolysis-inhibition assay uses fresh serum, it may more accurately reflect the protein's inhibitory effect. In contrast, our ELISA data primarily provide qualitative evidence of reduced binding affinity.

Here are possible explanations for the N29R. First, the ELISA experiment used commercial C9 protein extracted from human serum through multi-step purification, the final preserved and used C9 protein may have undergone slight denaturation, leading to undetected binding between C9 and mutant binder. Second, the mutant binder may also interact nonspecifically with other serum complement proteins, resulting in partial inhibition effects in hemolysis.

- *Fig 5E – adjust the x-axis to better read the scale.*

Thanks, we have adjusted the x-axis of **Figure 5e** to better read the scale (**p. 30**).

- *In the last results section, the final sentence is more appropriate for the discussion. “For emergency treatment of acute hemolysis, extending the therapeutic window by just a few minutes is crucial for saving patients’ lives and minimizing tissue damage.*

Thanks, we have adjusted this sentence to the discussion section (**p. 11, line 324-326**).

- *5D. why this approach of first allowing lysis and then add inhibitors. Repeat in the ‘classical way’.*

Thanks, the in vitro experiment shown in **Figure 5d** was designed to compare the inhibitory efficacy of the miniprotein binder with the C5 inhibitor eculizumab after complement activation had already been initiated for some time. The results of the classical inhibition assay, in which the inhibitors were pre-incubated with serum before being added to the erythrocyte-containing reaction system, are presented in **Figure 5a, 5b and 5f** (**p. 30**).

(7) [Discussion]

For discussion, consider expanding to include the following

- *Computation resources required to pursue such a study. It’s only briefly mentioned and much of the emerging software is also computationally expensive.*

Thanks, we have added discussion content on computational costs (**p. 9-10, line 284-291**).

- *Impact of depleting C9 on native state, chronic conditions, infection, etc.*

Thanks, we have added relevant discussion on the possible impact of blocking C9 membrane insertion (**p. 10-11, line 311-318**).

- *Kon rates are discussed but what about Koff?*

Thanks, we have added relevant description (**p. 10, line 294-295**).

- *Longterm treatment of binder?*

Thanks, we have added discussion on longterm treatment of binder (**p. 11, line 333-339**).

Reviewer #3:

Li et. al. present their design of mini-protein binders for complement C9, binders capable of inhibiting MAC formation. Using recently established protein design methods to find C9 binders (RFDiffusion, ProtMPNN, AlphaFold2) they discover a number of binders to monomeric C9, some with impressively high affinity, and solve the structure of a bound complex, which is very similar to the predicted design. The in vitro tests reveal that these new inhibitors have potent activity, and in many respects outform an approved antibody biologic. This is an experimentally solid, well-presented study that is an excellent addition to the growing number of studies that show the power diffusion methods in de novo protein design. Their argument about the value of early experimental analysis, and some limits of computational metrics, will be valuable to the field. And the new biologics found have clear biomedical potential.

I would recommend accept, I only have minor changes, mostly the clarify some statements:

- “and the inhibitors are kinetically disadvantaged.” What are “the” inhibitors here? Existing monoclonal antibodies? Please clarify.*

Thanks, we have clarified this statements (p. 2, line 67).

- “Despite this therapeutic rationale, C9-directed inhibitors remain rarely reported.” If they have been reported at all, references are needed.*

Thanks, we have added reported inhibitors in the main text (p. 2-3, line 80-84).

An early report showed that a commercially available anti-C9 monoclonal antibody X197 conferred inhibitory activity in a non-standard hemolysis-inhibition assay³. Separately, the O1-antigen of Klebsiella LPS was found to block C9 polymerization⁴. However, neither study evaluated the therapeutic potential of these inhibitors for hemolytic disorders.

- “Moreover, the relatively high serum concentration of C9 requires that C9 inhibitors have high affinity” Is this right, I would expect the opposite. If the concentration of target is high, then any binder with a Kd below this concentration will bind. High doses might be required, but not high affinity.*

Thanks, we have changed this statements (p. 3, line 86-88).

- “we can now generate binding proteins for any structural epitope” this could be too strong a statement. Indeed, in the same paragraphy it is claimed that “C9 remains challenging”*

Thanks, this sentence is indeed inappropriate, we have revised the relevant wording (**p. 3, line 90-92**).

- *“The soluble, monomeric crystal structure of mouse C9” Why not the human, or the predicted human structure (that is used later)?*

Thanks, we initially used murine complement C9 as the target, because the crystal structure of the soluble murine C9 monomer had been experimentally determined, whereas that of the human homolog had not. The experimentally resolved structure is more credible and, in principle, more accurate than a predicted human soluble C9 monomer model. We added descriptions in the main text (**p. 3, line 106-108**).

In addition, based on the membrane-inserted human C9 structure, it is inferred that the overall architecture is structurally conserved between human and murine C9. Since the initial computational screen aimed to obtain the best mini-protein binder scaffolds capable of binding and inhibition, we performed the initial calculations with the crystal structure of murine C9 and switched to human C9 for subsequent affinity maturation.

For the “sheep red blood cell hemolysis inhibition assay” the organism of the serum should be stated in the main text. As written, it could be mistaken for sheep serum.

Thanks, this sentence is indeed misleading, we have revised the relevant wording (**p. 5, line 145-147**).

- *“In the future, optimizing partial diffusion using complement C9 structures from different species will result in better performance binders.” What does “better” mean in this context? Able to bind more species, or bind human better?*

Thanks, this sentence is indeed misleading, we have revised the relevant wording (**p. 7, line 202-204**).

- *“were mutated to alanine to enhance diffraction”. Not obvious to me why this would work, and no reference given.*

Thanks, we have added relevant references in the main text (**p. 7, line 212-214**); alanine mutation of surface residues to promote crystal growth is a commonly used crystallographic strategy^{5, 6, 7, 8}. Alanine mutations enhance the probability of protein crystallization by reducing surface entropy, optimizing crystal contact interfaces, and minimizing hydration interference⁸.

References

1. Watson JL, *et al.* De novo design of protein structure and function with RFdiffusion. *Nature* **620**, 1089-1100 (2023).
2. Glögl M, *et al.* Target-conditioned diffusion generates potent TNFR superfamily antagonists and agonists. *Science (New York, NY)* **386**, 1154-1161 (2024).
3. Hatanaka M, Seya T, Yoden A, Fukamoto K, Semba T, Inai S. Analysis of C5b-8 binding sites in the C9 molecule using monoclonal antibodies: participation of two separate epitopes of C9 in C5b-8 binding. *Molecular immunology* **29**, 911-916 (1992).
4. Masson FM, *et al.* Klebsiella LPS O1-antigen prevents complement-mediated killing by inhibiting C9 polymerization. *Scientific reports* **14**, 20701 (2024).
5. Lu L, *et al.* De novo design of drug-binding proteins with predictable binding energy and specificity. *Science (New York, NY)* **384**, 106-112 (2024).
6. Yu B, *et al.* De novo design of light-responsive protein-protein interactions enables reversible formation of protein assemblies. *Nature chemistry* **17**, 1910-1919 (2025).
7. Yin W, *et al.* Crystal structure of the human 5-HT(1B) serotonin receptor bound to an inverse agonist. *Cell discovery* **4**, 12 (2018).
8. Derewenda ZS. The use of recombinant methods and molecular engineering in protein crystallization. *Methods (San Diego, Calif)* **34**, 354-363 (2004).